# The Role of Transarterial Embolization Plus Radiotherapy Compared to Radiotherapy or Transarterial Embolization Alone in the Management of Painful Bone Metastases: Results of a Systematic Review

**DOI:** 10.3390/cancers16244183

**Published:** 2024-12-15

**Authors:** Antonio Vizzuso, Matteo Renzulli, Valentina Lancellotta, Alessandro Posa, Patrizia Cornacchione, Bruno Fionda, Ciro Mazzarella, Davide De Leoni, Luca Tagliaferri, Emanuela Giampalma, Roberto Iezzi

**Affiliations:** 1Radiology Unit, “G.B. Morgagni” Hospital, AUSL Romagna, 47121 Forlì, Italy; matteo.renzulli@unibo.it (M.R.); emanuela.giampalma@auslromagna.it (E.G.); 2Department of Medical and Surgical Sciences, University of Bologna, 40124 Bologna, Italy; 3Department of Diagnostic Imaging, Oncological Radiotherapy and Hematology, University Polyclinic Foundation “A. Gemelli” IRCCS, UOC Oncological Radiotherapy, 00168 Roma, Italy; valentina.lancellotta@policlinicogemelli.it (V.L.); patrizia.cornacchione@policlinicogemelli.it (P.C.); bruno.fionda@policlinicogemelli.it (B.F.); ciro.mazzarella@policlinicogemelli.it (C.M.); luca.tagliaferri@policlinicogemelli.it (L.T.); 4Department of Diagnostic Imaging, Oncological Radiotherapy and Hematology, University Polyclinic Foundation “A. Gemelli” IRCCS, UOC General Diagnostic and Interventional Radiology, 00168 Roma, Italy; alessandro.posa@policlinicogemelli.it (A.P.); davide.deleoni01@icatt.it (D.D.L.); roberto.iezzi@policlinicogemelli.it (R.I.); 5Catholic University of the Sacred Heart, 00168 Roma, Italy

**Keywords:** embolization, radiotherapy, bon metastases, pain

## Abstract

This study examines the use of transarterial embolization (TAE) combined with radiotherapy (RT) for treating bone metastases, which can cause severe pain and complications. By combining TAE and RT, the researchers aimed to evaluate whether this approach improves disease control and pain relief more effectively than RT or TAE alone. A review of studies found that TAE plus RT may offer better local tumor control and quicker, longer-lasting pain relief than RT alone, though it is linked to mild, temporary side effects. Despite promising findings, the limited number of studies and data make the overall quality of evidence very low. More research is needed to confirm the benefits and risks of this combined treatment.

## 1. Introduction

Bone metastases (BM) are the most common malignant skeletal lesions [1,2]. Up to 65% of BM originate from breast tumors in woman and from prostate tumors in men; the remaining 35% of BM cases arise from kidney, thyroid, and lung cancers [1]. In total, 69% of BM occur in the vertebrae, followed by pelvic bones, long bones, and the skull [3]. BM significantly influence patients’ quality of life due to pain, pathological fractures, and compression of nearby structures such as nerves. Pain is the most common skeletal-related event (SRE) and may be the first symptom in approximately 20% of patients [4]. Current treatment modalities to manage pain and prevent fractures are medical therapy, surgery, and radiotherapy (RT); the latter considered the best evidence-based non-interventional treatment for BM-related pain, even though some neoplasms (renal tumor or melanoma) are radio-resistant [5]. Pain relief usually starts 1–2 weeks after treatment initiation, and is observed in less than 60% of patients, with recurrent symptoms in up to 50% of cases [5]. Additionally, retreatment is not recommended if the maximal radiation dose has already been delivered and no immediate bone consolidation is provided [5]. However, RT remains the most common treatment for BM [2,6]. In this setting, interventional radiology procedures may play an important and complementary role to manage BM [2,3,4,5,6,7]. Thermal ablation is an effective tool that can be performed with curative or palliative intent [8,9,10]. Furthermore, some studies proposed a combined approach using RT and percutaneous treatments showing better results in terms of pain relief and local control when compared to a single treatment modality, without a significant increase in morbidity [8,9,10]. Minimally invasive procedures, such as balloon kyphoplasty or percutaneous vertebroplasty, may be beneficial in patients who have vertebral metastases without neurological compromise but with persistent pain [11]. Transarterial embolization (TAE) is another tool in the hands of the interventional radiologist which can be applied in the case of hypervascular BM to reduce pain and achieve bone consolidation. A large series of patients who underwent embolization of BM reported >50% and 97% pain relief and analgesic consumption reduction, respectively [12,13]. Embolization and radiotherapy are both effective therapeutic tools for managing pain and controlling disease in the context of BM. However, their combined use is not common.

The present systematic review was performed to assess the efficacy and the safety of the combined treatment of TAE plus RT compared to TAE and RT used alone, for the treatment of painful BM in terms of clinical response (CR), local disease control (LC), and adverse events (AE).

## 2. Materials and Methods

### 2.1. Development of Clinical Question

The clinical question was developed based on the P.I.C.O. framework: population (P), intervention (I), comparison (C), and outcomes (O). The clinical question was for (P) in painful BM, is TAE plus RT (I) superior when compared with RT or TAE alone (C), in relation to the outcomes (O) of benefit and harm?

### 2.2. Search Strategy and Evidence Selection

This systematic review was conducted in accordance with the PRISMA guidelines [14]. The protocol is registered with INPLASY, INPLASY2024110073.

We performed a comprehensive literature search using PubMed, Scopus, WebOfScience, Medline Plus, and the Cochrane Library to identify full-text articles evaluating the efficacy and safety of TAE plus RT compared to TAE or RT alone in BM. The website ClinicalTrials.gov was searched for ongoing or recently completed trials, and PROSPERO was searched for ongoing or recently completed systematic reviews. Further research was manually conducted through a reference check from the selected articles. The studies were identified using the following medical subject headings (MeSH) and keywords: “embolization”, “radiotherapy”, and “bone metastases”. The search strategy was (“embolization” [MESH] OR “chemoembolization” [All fields]) AND (“radiotherapy” [MESH]) OR “radiotherapy” [All fields]) AND (“bones” [MESH]) OR “osseous” [All fields]) AND (“metastases” [MESH]) OR “metastases” [All fields]). The search was restricted to papers published in English. Articles were selected for a full-text review based on title and abstract. We analyzed only clinical studies reporting on patients with BM who underwent TAE and RT, alone and/or as combined therapy. Conference papers, surveys, letters, editorials, book chapters, case reports, and reviews were excluded. A time restriction (1990–2024) of the publication was considered. Studies were identified through a search process performed by three independent reviewers (MR, AV, and PC), and uncertainty regarding eligibility was resolved by a multidisciplinary committee (VL, AP, and BF). Eligible citations were retrieved for a full-text review. Outcomes of benefit and harm were defined. The multidisciplinary committee coordinated the project and performed the final independent check and the definitive approval of the review (AV, MR, CM, DDL, EG, LT, and RI). The GRADEpro Guideline Development Tool (GDT) (McMaster University, 2015) was used to create a Summary of Findings table in the Cochrane systematic reviews. The quality assessment showed high clinical and methodological heterogeneity and risks of bias in the included studies; therefore, meta-analysis outcomes were not reported, and quantitative analysis was possible only on CR rates.

### 2.3. Identification of Outcomes

Among outcomes of benefit, CR rates were based on the patient’s subjective pain score: complete pain relief (CpR) defined as complete freedom from pain without analgesics; partial pain relief (PpR) defined as reduced pain but still requiring analgesics; no pain relief (NpR). LC rates were based on imaging evaluation, defining a complete response as a ≥50% decrease in tumor size, a partial response as a <50% decrease in tumor size, a stable response was defined as no change in tumor size, whereas progressive disease was defined as an increase in tumor size. The identified outcome of harm included local and systemic AE.

### 2.4. Quality of Evidence Evaluation

Certainty of evidence for all selected outcomes was performed according to the GRADE approach, considering study limitations, imprecision, indirectness, inconsistency, and publication biases. The certainty level starts at a higher pre-specified level for randomized controlled trials, but levels of certainty can be downgraded if limitations in one of the above-mentioned domains are detected. Evidence was classified as having high, moderate, low, and very low levels of certainty.

### 2.5. Benefit/Harm Balance and Clinical Recommendation

Based on the summary of evidence, the following judgments on the benefit-to-risk ratio between intervention and comparison were stated: favorable, uncertain/favorable, uncertain, uncertain/unfavorable, and unfavorable (both for intervention and comparison). The strength of the recommendation is considered as strong positive, conditional positive, uncertain, conditional negative, or strong negative.

## 3. Results

### 3.1. Search Strategy Results and Details of the Identified Relevant Studies

In total, 64 potentially relevant studies were identified through the database searches after duplicates and title removal. These articles were screened at title and abstract level, and 56 of them were excluded. During the full-text evaluation of the remaining 8 papers, 3 were excluded due to the lack of a comparison between TAE and RT groups, whereas 2 were excluded due to the lack of a combined therapeutic group (TAE plus RT). Three original manuscripts were eventually selected for analysis (Figure 1).

### 3.2. Studies Characteristics

The systematic review of the three studies was performed on 74 patients (19 female, median age 62.9 years (range 36–82 years), with a total of 103 BM included. In all the studies, patients presented single and multiple bony metastases [15,16,17]. Study design characteristics, CR rates, follow-up, and LC and AE rates were recorded (Table 1). Median follow-up ranged between 0.5 and 103 months (median: 6 months) [15,16,17]. BM were from hepatocellular carcinoma (HCC), renal cell carcinoma (RCC), and differentiated thyroid cancer [15,16,17]. Local chemotherapy drug infusion was performed in addition to TAE in one study [16]. Gel-sponge particles, polyvinyl alcohol (PVA) particles, microspheres, and coils were the embolic agents used to achieve target arterial embolization [15,16,17].

In all the studies, patients presented single and multiple bony metastases.

### 3.3. Studies Description

Uemura et al. investigated the treatment of BM from HCC [15]. The authors evaluated pain relief and pain recurrence in 33 patients (34 BM) presenting with osseous pain. BM were divided into three treatment groups: RT alone (group A, n = 9), TAE plus RT (group B, n = 16), and TAE alone (group C, n = 9). In the cases of combined therapies, RT sessions were commenced 1–6 days after TAE. Each treatment was effective, achieving an initial CR of 89%, 94%, and 89% in group A (0% CpR, 89% PpR; 8/9 cases), group B (13% CpR, 81% PpR; 15/16 cases), and group C (22% CpR, 67% PpR; 8/9 cases), respectively. TAE provided more prompt pain relief (*p* < 0.01), with a mean time interval between the beginning of each treatment and the onset of initial pain relief of 15, 4.8, and 4.7 days for group A, B, and C, respectively. Mean follow-up was 182 days, with a range of 16–1977 days. The combined therapy of TAE plus RT provided a higher rate of permanent pain relief (*p* < 0.05); pain recurrence after initial pain relief was 20% for the combined group (group B), whereas it was 88% and 75% for groups A and C, respectively. Therefore, regarding the patients with CR, the persistent pain relief at maximum median follow-up (6 months) was 12% (1/8 cases), 80% (12/15), and 25% (2/8) for groups A, B, and C, respectively. Three BM underwent two consecutive TAE sessions at 10–17 days interval, for a total of 28 TAE performed with gel-sponge particles, PVA particles, and coils; of these, 26 (93%) determined a complete interruption of blood supply or a significant decrease in tumor vascularity at post-procedural angiography, without complications [15]. Heianna et al. published a retrospective study describing the therapeutic results in 25 patients with RCC (28 BM) [16]. The authors added a chemotherapy drug infusion to TAE to improve its efficacy. One group of patients received RT (16 patients, 17 BM), while the other received combined treatment (9 patients, 11 BM), with TAE performed after completion of RT session. Cisplatin (in patients with good renal function) and carboplatin (in patients with impaired renal function) were used as a local chemotherapy drug. Embolizing microparticles determined the disappearance of tumor contrast enhancement at post-procedural angiography. No patients received any systemic chemotherapy or bone-modifying agent during treatment. In terms of outcome, the authors considered the skeleton-related events (SRE), defined as local pain, pathological fractures, spinal cord compression, and need of bone surgery. None of the patients in the combined group suffered from SRE, while 53% of lesions in the RT monotherapy group presented with SRE. The 2-years SRE-free rates in the monotherapy and combination therapy groups were 42% and 100%, respectively (*p* = 0.009). Two patients in the monotherapy group died before completing the LC evaluation. LC rates were worse for RT alone than combined therapy (33% and 82%, respectively; *p* = 0.009), whereas median tumor reduction rates were -2% and 18%, respectively (*p* = 0.014). Although pre-treatment median tumor size in the combined therapy group was significantly larger than that in the monotherapy group (*p* = 0.005), LC and tumor reduction rates of combined therapy were significantly higher than those of RT alone [16]. The retrospective study of Eustatia-Rutten et al. evaluated the long-term effects of embolization in 16 patients with symptomatic BM of differentiated thyroid carcinoma [17]. All patients were treated by thyroidectomy followed by radioiodine ablation therapy. A total of 41 embolization procedures were performed, using PVA particles; additional therapies (radioiodine, RT, and/or surgery) were associated with 26 TAE; more specifically, RT of BM was performed before embolization in 6 patients (8 TAE), and after embolization in 6 patients (7 TAE). Technical success was obtained in all 41 TAE. Successful embolization was defined as an improvement in clinical symptoms for at least 1 month without tumor progression at radiological examination (CT or MRI); this definition, therefore, includes both CR and LC. No success rate differences were observed between TAE procedures that were or were not preceded or followed by RT (62% and 60%, respectively, *p* = 0.923), even though the success duration was highly variable. Overall median efficacy duration was 11 months (range 1–84), significantly differing between TAE with and without RT: 15.5 versus 7 months, respectively (*p* = 0.192) [17].

### 3.4. Outcomes of Benefit

In terms of CR, overall quantitative analysis using the inverse variance-weighted average method showed a RR of 0.15 (CI: 0.03–0.69) in favor of combined therapy compared to RT alone. On the contrary, the same quantitative analysis demonstrated no significant difference between the combined therapy and TAE alone (RR 0.91; CI: 0.51–1.63) in terms of CR.

LC was reported only by Heianna et al., with worse rates for RT alone than TAE plus RT (33% and 82%, respectively; *p* = 0.009) [16]. In the study by Eustatia-Rutten et al., LC was assimilated in the CR definition and, therefore, is difficult to extrapolate [17]. The Summary of Findings table for outcomes of benefit is reported in Table 2. The certainty of evidence was judged as “very low” for each outcome of benefit for the following reasons: indirectness for differences in outcome measures (surrogate outcomes), imprecision for sample size, and selection bias [15,16,17].

### 3.5. Outcomes of Harm

All three studies reported data on toxicity (Table 1). Uemura et al. reported that the overall number of complications spread across the three group; 7% (4/63) in the RT group, 60% (38/63) in the TAE plus RT group, and 33% (21/63) in the TAE group [15]. The most common complication was post-embolization syndrome (pain and fever) in the two groups in which TAE was performed. Heianna et al. described AE in the TAE group only [16]. Although intra-arterial chemotherapy agents are selectively released at the tumor site, which reduces the AE associated with systemic toxicity, leukopenia (grade 3) and thrombocytopenia were each observed in two patients (13%) for the RT group whereas in the TACE plus RT group, leukopenia (grade 3) and thrombocytopenia were described in three patients and one patient (44%), respectively. However, in all these patients, hematotoxicity decreased without intervention. In the study published by Eustatia-Rutten et al., AE were observed in two cases after embolization (5%); one developed a post-embolization syndrome and in one other patient, a contrast reaction occurred [17].

Concerning the AE, the quality assessment showed high clinical and methodological heterogeneity making quantitative synthesis inappropriate. The certainty of evidence was considered as “very low” for each outcome of harm for the following reasons: indirectness for differences in outcome measures (surrogate outcomes), imprecision for sample size, and selection bias [15,16,17].

### 3.6. Evidence to Decision Framework

In the BM setting, the proposed intervention (TAE/TACE plus RT) has proven effective in local tumor control and improving pain relief, as being faster and more durable compared to RT alone, whereas the combined therapy did not demonstrate significant results against embolization therapy alone. Moreover, TAE/TACE was associated with a higher incidence of side effects, but these were mild and transient.

### 3.7. Benefit/Harm Balance and Final Recommendation

The strength of the recommendation was low. Hence, the final recommendation was “In patients with BM metastases, the combination of TAE/TACE plus RT should be evaluated on an individual basis through discussion by a multidisciplinary team”.

## 4. Discussion

Pain is a subjective experience, and its intensity cannot be predicted by the tumor type, size, and number of metastases or bone involvement. Cancer-induced bone pain is multifactorial and involves several mechanisms as it is the result of various interactions between tumor cells, bone cells, activated inflammatory cells, and bone-innervating neurons. It includes inflammatory and neuropathic processes, which are modified at the level of peripheral tissues and nerves, as well as at higher levels of the nervous system (the spinal cord and brain) [1,18,19]. Therefore, these observations warrant an individualized approach to each patient’s painful BM and its management should be multimodal (pharmacological and non-pharmacological), including causal anticancer and symptomatic analgesic treatment [20]. RT and TAE are casual treatment of the pain as they reduce the tumor mass, perfusion, and local tissue infiltration [1]. Previous studies indicated that RT may destroy arterioles and inhibit capillary growth in the inner parts of tumors whereas larger tumor-feeding arteries remain intact. Therefore, tumor regrowth, after RT, starts from the outermost regions of the tumor, where vascularity is greatest, suggesting that a complete recovery of tumor vascularity and regrowth of arterioles is expected to begin soon after treatment is stopped [16,21,22,23]. Embolization occluded larger tumor-feeding vessels (≥0.5 mm in diameter) resulting in an immediate reduction in tumor vascularity, which results in the prompt decompression of the periosteum supplied by abundant nerves and a reduction in tumor bulk. However, oxygen deprivation after embolization is a major stimulus for tumor-induced neovascularization and tumor relapse although irradiation may prevent this effect [23,24]. From a pathophysiological point of view, RT and embolization have different targets and it is expected that they could have an additive and long-lasting clinical outcome due to the synergistic effect of destruction of arterioles and the occlusion of larger feeding arteries. In our systematic review, we highlighted that TAE plus RT were superior to RT alone in achieving pain relief and, in the study with concomitant local infusion of chemotherapy agent, the combined therapy was significantly superior to RT alone [16]. TACE plus RT has also demonstrated a significantly superior objective response probably explained because embolization of hypervascular bone lesions slows blood flow down, delaying clearance of the infused anti-cancer agents, and increasing the duration of exposure to the anti-cancer agent allowing a further additional effect [25]. Thus, local chemotherapy infusion reveals an advantage because of a combined ischemic and cytotoxic effect [26].

The most evident, superior effect of TACE compared to TAE could have also been justified by the used embolic agent. In fact, Heianna et al. used microspheres, which offer several advantages over gelatin sponge and PVA particles, leading to a more complete and deeper vascular occlusion [16]. Gelatin sponge is conventionally regarded as a temporary embolic agent; it produces thrombus formation due to a necrotizing arteritis and foreign body reaction, but these inflammatory changes persist for approximately up to 45 days. Although not biodegradable, and thus, considered a permanent embolization material, PVA particles also provide angiogenesis-mediated recanalization of embolized vessels within the organized thrombus, resulting in thrombus resorption. PVA particles have a high coefficient of friction and can aggregate, leading to proximal rather than distal embolization, as well as using coils [27]. In this latter scenario, and particularly without RT association, the undamaged inner parts of the tumors can induce vessel recruitment and tumor recurrence, with a mechanism that is similar to an arteriovenous malformation’s nidus managed with proximal feeding artery embolization only [28]. To date, liquid agents, such as N-butyl cyanoacrylate, are more useful, as they allow fast, permanent occlusion, and deep penetration within the tumor [24].

In line with the aforementioned biological effects and anatomical targets, this systemic review has shown that the therapeutic effects of embolization (alone or in combined therapy) appeared more rapidly based on the reduction in tumor vascularity that helps prompt decompression on the periosteum containing the nerves and, in combination with RT, provided a higher rate of permanent pain relief due to the synergic effect that prevent rapid relapse.

The technical success of embolization, defined as occlusion of all tumor-feeding artery with 80% or greater elimination of the tumor pathologic contrast enhancement visible on conventional angiography compared with the initial diagnostic angiogram, was reported in two studies with an overall success rate of 97% (range 93–100%) [12,15,17].

A combination of TAE/TACE plus RT was found to have a tolerable toxicity profile. Pain and fever are part of the so-called post-embolization syndrome, which is the most common complication of TAE, even if temporary; this explain the higher complication rate reported in the embolization groups [15,16,17]. Ischemic pain at the embolization site, paresthesia, skin breakdown, and sub-cutaneous necrosis are other common AE after bone embolization; however, these AE were not reported in the included studies [12]. In the study that performed TACE, there was no grade ≥4 blood toxicity [16]. However, grade-3 leukopenia and thrombocytopenia occurred and, even though no patient required intervention, the potential for serious chemo toxicity should always be considered when administering anticancer agents, and their indication should be carefully accounted.

All included studies were performed retrospectively, rendering them prone to multiple types of bias. Limitations of our systematic review include the small number of studies and patients enrolled with heterogeneity of demographic characteristics, primary tumor site and etiology of BM, RT dose delivered, previous treatment, timing of additional therapy, and techniques of embolotherapy. The follow-up period was short both for outcomes of benefit and harm, although a longer follow-up is unlikely as patients with BM have a short life expectancy due to their underlying disease. Of the three included studies, only one reported the LC and, regarding the pain relief, the same study extrapolated it from SRE [16]. Furthermore, the study by Heianna et al. investigated the outcome of benefit regarding the hypervascular RT-resistant metastatic RCC, with a good clinical and objective response [16]. Although the same authors published another retrospective study in which TACE and RT alone were compared in different BM (gastrointestinal, urogenital, head and neck, sarcoma, lung, breast, thymus) and showing high pain relief rates after embolization than RT (92% versus 57%, respectively; *p* = 0.006), more durable pain relief (51% versus 30% at 6 months follow-up, respectively; *p* = 0.08), and faster pain relief (92% of patients experienced pain relief the day after), we believe that new research is needed to investigate the effective role of the combined approach in the setting of BM in which RT alone has a clear and well-defined therapeutic role [26]. This systematic review has shown, prior to any clinical outcomes, that there is a lack of robust data to extrapolate recommendations with high levels of evidence in this field. However, in daily clinical practice, these patients need to be treated. Therefore, our paper, in the absence of randomized clinical trials or prospective studies, allows us to state that, due to the lack of strong evidence, each patient should be discussed in a multidisciplinary meeting with the involvement of radiation oncologists and interventional radiologists, who can jointly assess the current best possible approach for these patients with bone metastases: combined radiotherapy and endovascular treatment. Our hope is also that this paper, by highlighting the absence of scientific robust data, will encourage the scientific community to plan studies with designs specifically aimed at answering the many unresolved questions, possibly through data sharing across different centers.

## 5. Conclusions

The lack of high-quality evidence and the aforementioned biases hinder an accurate definition of the effective role of embolization plus RT in this setting. Despite these limitations, the current systematic review provides an orientation and suggests that combined therapy with TAE and RT is an interesting treatment for BM. TACE/TAE plus RT results in better CR and LC compared with RT alone. Embolization demonstrates more rapid pain relief and, in combined therapy, a higher rate of permanent pain relief compared to RT alone.

## Figures and Tables

**Figure 1 cancers-16-04183-f001:**
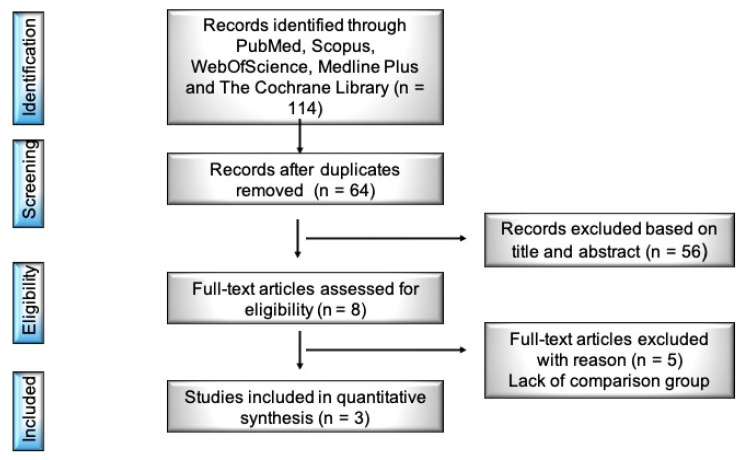
Study selection process.

**Table 1 cancers-16-04183-t001:** Study design characteristics, CR, follow-up, LC and AE.

Author	Study	Primary Tumor	Therapy	N° Patients	N° Lesions	Median Follow-Up (Months)	Technical Success (%)	CR (%)	Onset of Initial Pain Relief (Days)	CR at the Maximum Follow-up Reported (%)	LC (%)	AE (%)
Uemura et al. (2001) [15]	retrospective	HCC	RT + TAE	33	16	6 (0.5–66)	26/28 (93)	15/16 (94)	4.8	at 6 month 12/15 (80)	N/A	38/63 (60)
			TAE		9			8/9 (89)	4.7	at 6 month 2/8 (25)	N/A	21/63 (33)
			RT		9		N/A	8/9 (89)	15	at 6 month 1/8 (12)	N/A	4/63 (7)
Heianna et al. (2019) [16]	retrospective	RCC	RT + TACE	9	11	9 (2–29)	N/A	11/11 (100)	N/A	at 24 month (100)	9/11 (82)	4/9 (44)
			RT	16	17	10 (1–38)	N/A	8/17 (47)	N/A	at 24 month (42)	5/15 (33)	2/16 (13)
Eustatia-Rutten et al. (2003) [17]	retrospective	Thyroid Carcinoma	RT + TAE	16	26	(2–103)	41/41 (100)	16/26 (62)	N/A	N/A	N/A	2/41 (5)
			TAE		15			6/9 (60)	N/A	N/A	N/A	

Abbreviations: RT: radiotherapy; TAE: transarterial embolization; TACE: transarterial chemoembolization; CR: clinical response; LC: local control; AE: adverse events; HCC: hepatocellular carcinoma; RCC: renal cell carcinoma, N/A: not applicable.

**Table 2 cancers-16-04183-t002:** Summary of Findings table for outcomes of benefit.

Certainty Assessment							№ of Patients		Effect		Certainty
№ of Studies	Study Design	Risk of Bias	Inconsistency	Indirectness	Imprecision	Other Considerations	TAE/TACE + RT	TAE or RT Alone	Relative (95% CI)	Absolute (95% CI)	
**CR (Uemura A, Eur Radiol. 2001) (follow up: median 6 months; evaluated with: events; primary tumor site: HCC)**											
1	Observational	Serious ^a^	Not serious	Not serious	Serious ^c^	None	CR (CpR + PpR) 94% in TAE + RT group vs. 89% in the TAE alone group vs. 89% in RT alone group. TAE + RT group vs. RT alone group: RR 0.53 (0.04–7.44) TAE + RT group vs. TAE alone group: RR 0.53 (0.04–7.44)				- Very low
							6-months CR (CpR + PpR) 80% in TAE + RT group vs. 25% in the TAE alone group vs. 12% in RT alone group (*p* < 0.05) The earliest onset of initial CR after treatment was 4.8 days for TAE + RT group vs. 4.7 days for TAE alone group vs. 15 days for RT alone group (*p* < 0.01)				
**SRE (Heianna J, Eur Radiol. 2020) (median follow up: 9 months in TACE + RT group, 10 months in RT group; evaluated with: event; primary tumor site: RCC)**											
1	Observational	Serious ^a^	Not serious	Serious ^b^	Serious ^c^	None	SRE-free rate 100% in the TACE + RT group vs. 47% in the RT alone group TACE + RT group vs. RT alone group: RR 0.08 (0.01–0.52)				- Very low
							2 years SRE-free rate 100% in the TACE + RT group vs. 42% in RT alone group (*p* = 0.009)				
**LC (Heianna J, Eur Radiol. 2020) (median follow up: 9 months in TACE + RT group, 10 months in RT group; evaluated with: event; primary tumor site: RCC)**											
1	Observational	Serious ^a^	Not serious	Not serious	Serious ^c^	None	LC 82% in the TACE + RT group vs. 33% in RT alone group (*p* = 0.009)				- Very low
**CR (Eustatia-Rutten CF, J Clin Endocrinol Metab. 2003) (follow up: range 2-103 months; evaluated with: event; primary tumor site: thyroid)**											
1	Observational	Serious ^a^	Not serious	Serious ^b^	Serious ^c^	None	CR 62% in TAE + RT group vs. 60% in TAE alone group (*p* = 0.923) TAE + RT group vs. TAE alone group: RR 0.94 (0.52–1.70)				- Very low
							The median duration of CR was 15.5 months in TAE+RT vs. 7 months in TAE alone group (*p* = 0.192)				

Abbreviations: RT: radiotherapy; TAE: transarterial embolization; TACE: transarterial chemoembolization; CR: Clinical response; CpR: complete pain relief; PpR: Partial pain relief; LC: local control; SRE: Skeleton related events; HCC: hepatocellular carcinoma; RCC: renal cell carcinoma. CI: Confidence interval. a. Selection bias. b. Surrogate outcome. c. Small population.

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
