# Peer review of "The Role of Transarterial Embolization Plus Radiotherapy Compared to Radiotherapy or Transarterial Embolization Alone in the Management of Painful Bone Metastases: Results of a Systematic Review"

_cancers, 2024, doi:10.3390/cancers16244183_

Round 1
Reviewer 1 Report
Comments and Suggestions for Authors
Compliments to the authors for doing a systematic review on the role of transarterial embolization plus radiotherapy compared to radiotherapy or transarterial embolization alone in the management of painful bone metastases.
This was a very interesting reading and a huge learning experience.
Few small questions:
1. Did the studies select patients with multiple bony metastasis?
2. Instead of saying that all of treated lesions in the combined group were not associated with SRE, better to say none of the patients in combined group suffered from SRE.
3. You have said in Discussion that the technical success of embolization is defined as occlusion of all tumor-feeding artery with a complete disappearance of tumor staining. What does tumor staining mean?
Reviewer 2 Report
Comments and Suggestions for Authors
The authors describe their systematical review on the efficacy and safety of combining transarterial embolization (TAE) with radiotherapy (RT) for managing bone metastases, assessing clinical response (CR) and local control. A quantitative analysis of CR rates showed a relative risk of 0.15 favoring TAE plus RT over RT alone, while no significant differences were observed between TAE plus RT and TAE alone. The combined TAE and RT approach demonstrated effectiveness in local tumor control and produced faster, longer-lasting pain relief than RT alone, although TAE was associated with a mild, transient increase in side effects. It was concluded that while TAE plus RT shows potential benefit and acceptable toxicity, the current evidence is of low quality. This is an interesting study. Appropriate methodology has been employed and the conclusions appear to be justified based on the data at hand. While the authors describe some limitations of their work, I have some recommendations for consideration.
1. Introduction. Please provide a stronger rationale as to why this study was undertaken as well as a clear hypothesis to be tested in the study.
2. Results/Discussion. It is not clear if the analyses and responsiveness to the treatment have taken sex into account, i.e. breast cancer vs. prostate cancer, it would be interesting.
3. Discussion. In this reviewers opinion, this systematic review is preliminary. Please comment. Also, the authors need to elaborate and emphasize on the novelty aspect of their work.
4. Discussion. The authors need to further highlight the clinical applicability of their findings.
Reviewer 3 Report
Comments and Suggestions for Authors
Present study compares the alone against combinational approach of radiotherapy with transarterial embolization in bone metastasis for pain. The data from 74 patients was studied from three published studies with major parameters including pain relief and tumor control. The findings from the studies showed that transarterial embolization in combination with radiotherapy to the patients with bone metastasis provided long duration of pain relief. Though the idea is quite good and may have promising outcomes however, high bias risks, small number of patients and varied treatment methods exist.
I feel that these studies may not be recommended as due to small number of patients the reproducibility of the findings may be different particularly when there are several kind of variations in the bone metastasis itself.
Some of the other observations are:
Retrospective studies are quite prone towards biasness.
There could be several variations in the findings with respect to patient demographics, bone metastsais type, Radiotherapy dosing, procedure used for Transarterial embolization
The authors focused on only three studies, which itself is quite low in number.
There are some conflicts in the outcome too as only one study concluded the clinical response in comparison to control.
The authors are suggested to add more relevant data and studies for broad comparison.
Round 2
Reviewer 3 Report
Comments and Suggestions for Authors
The authors have addressed my comments, and I am in agreement for the acceptance of this version of manuscript for publication